# Two Hawks with One Arrow: A Review on Bifunctional Scaffolds for Photothermal Therapy and Bone Regeneration

**DOI:** 10.3390/nano13030551

**Published:** 2023-01-29

**Authors:** Yulong Zhang, Xueyu Liu, Chongrui Geng, Hongyu Shen, Qiupeng Zhang, Yuqing Miao, Jingxiang Wu, Ruizhuo Ouyang, Shuang Zhou

**Affiliations:** 1Institute of Bismuth and Rhenium Science, University of Shanghai for Science and Technology, Shanghai 200093, China; 2Shanghai Chest Hospital, School of Medicine, Shanghai Jiao Tong University, Shanghai 200030, China; 3Cancer Institute, School of Medicine, Tongji University, Shanghai 200092, China

**Keywords:** bifunctional scaffolds, PTT, PTAs, bone tumor, bone regeneration

## Abstract

Despite the significant improvement in the survival rate of cancer patients, the total cure of bone cancer is still a knotty clinical challenge. Traditional surgical resectionof bone tumors is less than satisfactory, which inevitably results in bone defects and the inevitable residual tumor cells. For the purpose of realizing minimal invasiveness and local curative effects, photothermal therapy (PTT) under the irradiation of near-infrared light has made extensive progress in ablating tumors, and various photothermal therapeutic agents (PTAs) for the treatment of bone tumors have thus been reported in the past few years, has and have tended to focus on osteogenic bio-scaffolds modified with PTAs in order to break through the limitation that PTT lacks, osteogenic capacity. These so-called bifunctional scaffolds simultaneously ablate bone tumors and generate new tissues at the bone defects. This review summarizes the recent application progress of various bifunctional scaffolds and puts forward some practical constraints and future perspectives on bifunctional scaffolds for tumor therapy and bone regeneration: two hawks with one arrow.

## 1. Introduction

The total cure of bone cancer is still a knotty clinical challenge, though the survival rate of cancer patients has improved significantly with the progress of modern technology [1]. Bone cancer usually refers to malignant tumors of bone that are classified into primary and metastatic bone tumors according to the origin of the tumor. Primary bone tumors, especially osteosarcoma, are common in adolescents and young adults [2]. Early clinical signs of primary bone tumors are inconspicuous; therefore, the tumor has often grown sufficiently before the patient seeks medical attention [3]. Because of the suitable living environment in the bone marrow, the skeleton is conducive to tumor metastasis, which occurs in approximately 80% of breast or prostate cancers. Metastatic bone tumors are more common in middle-aged and older patients, which used to be considered advanced tumors and incurable [4]. Both bone tumors lead to unbearable pain associated with serious skeletal lesions, causing great distress to patients [5,6].

It is urgently required to develop noninvasive and efficient novel clinical treatments since conventional surgical resectioning of bone tumors is usually characteristic of inevitable residual tumor cells and large area bone defects [7]. Photothermal therapy (PTT) was developed by using photothermal therapeutic agents (PTAs), and generates the precisely controlled hyperthermia generated under near-infrared (NIR) light irradiation, which can efficiently ablate malignant tumors, thus providing a new minimally-invasive local therapy [8]. To date, diverse PTAs have been designed for PTT, including Au-based nanomaterials, carbon-based nanocomposites, metal chalcogenide nanomaterials, organic small molecules, and so on [9,10,11,12,13,14,15,16]. The therapeutic effects of both PTT and the combined therapies have been well validated by in vitro and vivo experiments. Such approaches could lend effective protection to bones from destruction by tumor cells; however, they have thus far failed to regenerate bones that were already damaged.

Bone is essentially a mineralized conjunctival tissue, chiefly made up of inorganic and organic components. The inorganic mineral in bones mainly consists of nanohydroxyapatite (nHA, Ca_10_(OH)_2_(PO_4_)_6_) and some trace elements [17,18]. Collagen (mainly type I) with osteoconductivity is the main organic component of the bone and has been considered a good template for mineralization [19]. To the present, utilizing bone grafts is the main careful method of repairing bone defects, including autograft, homograft, and heterograft; however, it usually comes with insufficient donors and high surgical risks [20]. The use of bioactive scaffolds is taken into account as a workable alternative to bone grafting because it perfectly avoids the above problems and has attracted worldwide attention in bone tissue engineering (BTE) in the past decade [21]. When implanting scaffolds into the bone defect sites, the cells, such as bone mesenchymal stem cells (BMSCs), or the nutrients and molecules required by the cells, are transported to the inner part of the scaffolds to promote cell growth, vascularization, and ultimately promote new bone formation [22]. Appropriate pore size and porosity are required for scaffolds which may affect ossification and angiogenesis, as well as mechanical and degradation properties that are necessary for osteogenesis [23]. These scaffolds are mainly divided into inorganic bioceramics and organic polymers, bringing demonstrated possibilities for accomplishing ideal osteogenesis.

More recently, a new type of biomaterial was reported to be able to present dual functions of significant tumor ablation and osteogenesis simultaneously via integrating PTAs into scaffolds [24,25], known as the strategy of “two hawks with one arrow” (Figure 1). Given the excellent performance of bifunctional scaffolds, this review focuses on the progress of various bifunctional scaffolds in the treatment of bone tumors and provides some practical constraints and future perspectives.

## 2. Bioceramic Scaffolds

Bioceramics refers to a class of ceramic materials used for specific biological or physiological functions, which means that ceramic materials are directly used in the human body or related to the human body in biological, medical, biochemical, etc. applications [26]. Calcium phosphate and bioactive glass (BG) are the two common types of bioceramics in BTE. Because of their bioactivity, ions can be continuously released as the materials degrade to promote osteoblast proliferation and differentiation, eventually leading to the formation of new bone tissue.

### 2.1. Calcium Phosphate Scaffolds

The use of phosphates to repair and rebuild the damaged parts of bone has a long history in healthcare, dating back to the 1770s [27]. The most widely studied calcium phosphate bioceramics include nHA and tricalcium phosphate (TCP).

#### 2.1.1. nHA

As the main inorganic component in human bones, nHA has inherent biocompatibility and osteoconductivity; however, the poor mechanical strength and slow degradation hindered its wide application [28]. These problems are usually solved by compounding nHA with organic polymers such as chitosan (CS) and gelatin (Gel) [29]. Lu et al. succeeded in doping carbon dots (CD) into CS/nHA composite scaffold by facile physical mixing, which not only endowed the scaffold with excellent photothermal and antibacterial properties but also enhanced the osteogenic capacity (Figure 1) [30]. The scaffolds showed a porous structure conducive to the adhesion of rat BMSCs. The doping of CDs not only up-regulated the expression of osteogenic genes, but also significantly promoted the formation of vascularized bone tissue to further promote bone formation. After being implanted into the tumor in osteosarcoma-bearing mice, the scaffolds effectively inhibited tumor growth in vivo under NIR irradiation of 808 nm (1 W/cm^2^). In addition, NIR radiation further enhanced the antibacterial activity of the scaffolds, and significantly killed the clinical bacteria *Staphylococcus aureus* and *Escherichia coli*.

#### 2.1.2. TCP

The chemical composition of TCP is similar to that of nHA. According to different lattices, TCP is divided into α-TCP and β-TCP, and the latter is more used in BTE because of the appropriate degradation rate [31]. The initially reported bifunctional scaffolds were graphene oxide (GO)-modified β-TCP scaffolds [24]. GO with high photothermal conversion efficiency (PCE) and satisfactory cytocompatibility could accelerate cell proliferation and differentiation [32]. However, its therapeutic efficacy was limited by potential toxicity and long-term safety [33]. Carbon aerogel (CA) exhibiting multiple unique properties such as large surface area, ultra-low density, and excellent thermal stability might be a substitute for other C-based PTAs. Dong et al. coated CA on β-TCP scaffolds for the treatment of osteosarcoma [34]. Under the NIR irradiation of an 808-nm laser (1 W/cm^2^), the scaffolds generated enough heat to ablate osteosarcoma cells, with a cancer inhibition rate of up to 81.85%. Scaffolds could enhance the attachment of bone marrow stromal cells and regulate subsequent cell behavior without irradiation. Due to the extra roughness and higher specific surface area provided by the CA coating surface, protein recruitment capacity was promoted, which accelerated the adhesion and osteogenic differentiation of BMSCs, thereby stimulating new bone formation.

Compared with nHA, β-TCP is more easily absorbed and replaced by new bone; however, it faces the problem of weak mechanical strength as well [35]. Dang et al. utilized the method of surface modification to enhance the mechanical strength of β-TCP scaffolds [36]. After coating with LaB_6_ and poly(d,l-lactide) (PDLLA), the average value of compressive strength of scaffolds was doubled and increased with the coating time. As well as releasing bioactive Ca^2+^ and PO_4_^3−^, the scaffolds could also release La^3+^ and BO_3_^3−^, which have been proven to induce the expression of osteogenesis-related genes [37,38]. Meanwhile, in vitro experiments show that tumor cells could be effectively eliminated under the irradiation of a NIR light (808 nm, 0.7 W/cm^2^) due to the unique photothermal properties and effects of LaB_6_. In addition, coated with PDLLA as a medium, scaffolds could be loaded with the antitumor drug doxorubicin (DOX) and TiN microparticles (Figure 2) or hemin particles (Figure 3) as PTAs to achieve the synergistic effect of chemotherapy and PTT on osteosarcoma [39,40].

Metal-organic frameworks (MOFs) are promising candidates for biomedical materials due to their excellent surface properties, biocompatibility, and multiple tunability [41]. In recent years, MOFs as PTAs have been designed for tumor therapy [42]. Dang et al. coated Cu-coordinated tetrakis (4-carboxyphenyl) porphyrin on the surface of β-TCP to fabricate scaffolds with photothermal and osteogenic properties [43]. Specifically, 90% of tumor cells could be killed after being irradiated for 10 min by a NIR light of 808 nm with a power density of 1.0 W/cm^2^. It is worth noting that Cu^2+^ could promote angiogenesis, which is a necessary factor to promote new bone formation [44]. After implantation into the rabbit bone defect, the scaffolds showed good bone formation biological activity and slow degradation, and the formation of new bone-related tissues could be observed, indicating that the scaffolds promoted the formation of new bone.

Ma et al. developed bifunctional scaffolds by coating 5% Cu-containing- mesoporous silica nanosphere (MSN) on the surface of β-TCP by spin coating [45]. The scaffolds effectively obliterated tumor cells and favored osteogenic differentiation in vitro. Unfortunately, the in vivo tumor ablation and bone regeneration capabilities of the scaffolds have yet to be validated. Furthermore, the drug-carrying capacity of MSN failed to be exploited for adjuvant cancer therapy.

In order to simultaneously eradicate tumors and repair tumor-related bone defects, Lin et al. constructed FeMg-NPs/TCP / Poly (lactic-co-glycolic acid) (PLGA) scaffolds based on PDA nanoparticles containing Fe^3+^ and Mg^2+^. The chemodynamic therapy (CDT) produced by Fe^3+^ works synergistically with PDA-induced PTT, resulting in a large number of tumor cell death and severe necrosis of tumor tissue [46]. The addition of Mg^2+^ enhanced the differentiation of osteoblasts and effectively repaired bone defects in vivo.

### 2.2. Silicate Scaffolds

#### 2.2.1. CaSiO_3_

Calcium silicate (CaSiO_3_) is the simplest silicate bioceramic. Fe-CaSiO_3_ composite scaffolds capable of synergistic therapy were fabricated via 3D printing (3DP) technology [47]. On the one hand, PTT could be provided by localized surface plasmon resonance of Fe. On the other hand, the Fenton reaction could be activated by the release of Fe^2+^, which promoted the decomposition of H_2_O_2_ to generate reactive oxygen species (ROS) to kill tumor cells (Figure 4). The synergistic effect of PTT and ROS increased the mortality rate of tumor cells to 91.4% in vitro. Moreover, the scaffolds exhibited a compressive strength of 126 MPa, which is within the compressive strength range of dense human bones, providing adequate mechanical support for cortical bone defects. In addition, when testing the osteogenic ability of the scaffolds in vivo, rabbits were irradiated by the NIR light of 808 nm at 0.8 W/cm^2^ for 10 min, and no adverse effect was found of the short-term PTT on long-term bone regeneration.

Fu et al. combined the 3DP and polymer-derived ceramics (PDCs) strategies to simply and efficiently fabricate free carbon-embedding larnite (β-CaSiO_3_) scaffolds [48]. The obtained scaffolds showed excellent photothermal and bone regeneration ability. After being irradiated with NIR light (808 nm, 1.25 W/cm^2^), the cellular activity of osteosarcoma cultured on larnite/C-3 scaffolds (MNNG/HOS) was only about 5%. It is worth noting that the introduction of free C could enhance the bone regeneration ability of larnite scaffolds. Specifically, the new bone mineral density (BMD) and bone volume/tissue volume ratio (BV/TV) of composite scaffolds (0.41 g/cm^3^, 40%) were higher than those of larnite scaffolds (0.3 g/cm^3^, 35%).

#### 2.2.2. Bioactive Glass (BG)

BG is generally a multi-component system with silicate as the main component, and some BGs based on borate or borosilicate components have also been applied in BTE [49]. The earliest BG was developed by Professor Hench in the early 1970s based on the SiO_2_-P_2_O_5_-CaO-Na_2_O quaternary system. In recent years, researchers have made BGs multifunctional to meet human metabolic needs by adding bioactive elements [50].

Cu-based chalcogenides, as a new class of efficient PTAs, have the advantages of convenient synthesis, strong absorption in the NIR region, and high PCE [9]. Dang et al. combined the photothermal properties of semiconducting CuFeSe_2_ with the osteogenic activity of BG to prepare bifunctional scaffolds [51]. PCE up to 82% allowed the scaffolds to heat up rapidly under 808-nm NIR laser irradiation at a power density of 0.5 W/cm^2^ to induce 96% of tumor cell apoptosis in vivo. The ions of Si, P, Ca, Cu, Fe, and Se that are released during scaffold degradation could work together to promote osteogenesis (Figure 5). Because Cu can induce Fenton reaction, Cu-based chalcogenides such as CuS have been developed for multimodal tumor therapy [52,53,54]. Unfortunately, at present, few of them have been combined with scaffolds for synergistic tumor therapy and bone regeneration.

BG scaffolds doped with Cu, Fe, Mn, and Co elements were prepared by 3DP technology, and the photothermal properties and osteogenic activities of these scaffolds were systematically investigated [55]. In addition to inhibiting tumor growth in vivo, Fe- or Mn-doped scaffolds also promoted the adhesion of rabbit bone mesenchymal stem cells (BMSCs), and the released Fe^3+^ or Mn^2+^ significantly stimulated the osteogenic differentiation of osteoblast. Cu-doped scaffolds, although owning the most excellent anti-tumor ability, were potentially cytotoxic. Most inorganic PTAs are plagued by this problem during the application and the preparation cost is relatively high because their constituent elements are mostly heavy metals and rare metals [56].

2D black phosphorus (BP) nanosheets with excellent photothermal properties are easily oxidized and degraded in aqueous media to become phosphate and phosphonate [57]. Based on this feature, Yang et al. prepared BP-reinforced 3D-Printed Scaffolds, and the PO_4_^3−^ generated by the degradation of BP could extract Ca^2+^ from the ambient physiological microenvironment for osteogenesis [58]. Quantitative analysis based on confocal fluorescence images showed that the introduction of BP enhanced the osteogenesis rate by 3.7-fold. When the scaffolds were implanted into the tumor center for PTT, the tumor tissue temperature rose sharply from 30 °C to 55 °C within 1 min and 58 °C within 5 min (808 nm, 1 W/cm^2^), and tumors in mice were eradicated and no longer reoccurred within 14 days.

Tremendous attention has been paid to organic PTAs due to their superior biodegradability, low toxicity, and structural diversity [59]. PDA with simple preparation and good biocompatibility has been widely studied in the fields of drug delivery and imaging as well as tumor therapy or diagnostic methods. Strong adhesion to almost all substrates makes PDA easy to combine with nanomaterials, greatly expanding their applications [12,60,61]. For the first time, Ma et al. applied organic PDA as PTAs into bifunctional scaffolds [25]. Saos2 cells and MDA-MB-231 cells were effectively killed even with NIR laser irradiation at an ultra-low laser power density (808 nm, 0.3 W/cm^2^). The absence of significant cell death in the control group demonstrated the excellent cytocompatibility of the scaffolds (Figure 6).

To date, a great deal of effort has been devoted to exploring organic PTAs such as cyanine dyes, porphyrin derivatives, and D-A type conjugated polymers [62,63]. Unlike inorganic PTAs, the lengthy synthetic steps of organic compounds are a familiar problem in the design of organic PTAs [64,65]. Inspired by the undemanding and economical manufacturing methods and prominent characteristics of organic cocrystals, Xiang et al. prepared NIR-absorbing organic cocrystalmodified BG scaffolds for NIR-activated photonic osteosarcoma hyperthermia and enhanced bone regeneration [66]. Based on a precise simulation of the band gap between the electron donor and acceptor of organic cocrystals, the scaffold showed outstanding PCE. The temperature of tumor sites administrated with scaffolds raised drastically from 32 to 53 °C within 2 min after exposure to the NIR laser (808 nm, 1 w/cm^2^) illumination. The tumors of mice were completely eradicated without further recurrence after PTT. More importantly, bifunctional scaffolds demonstrated enhanced abilities to stimulate osteogenic differentiation and angiogenesis, ultimately promoting the formation of new bone.

MXenes mainly include transition metal carbides and nitrides, in which “M” denotes transition metal atoms, “X” is C or N, and “ene” represents an ultrathin 2D structure. With fascinating physicochemical properties, MXenes have been explored in PTT as a new class of 2D materials [67]. Moreover, MXenes have been proven to show good biocompatibility and bone regeneration activity in vitro and vivo [68]. Employing the direct solution-soaking method, Pan et al. rationally integrated 2D Ti_3_C_2_ MXenes with BG for the first time to construct bifunctional scaffolds for bone cancer treatment (Figure 7) [69].

Most of the PTAs introduced before only absorbed the first near-infrared (NIR-I, 750–1000 nm) light with low maximum permissible exposure (MPE, 0.33 W cm^−2^ for 808 nm) and weak tissue penetration ability, which is unfavorable for practical clinical applications [70,71]. In contrast, the second near-infrared (NIR-II, 1000–1700 nm) light penetrates deeper into biological tissue and shows larger MPE (1.0 W cm^−2^ for 1064 nm), which is more suitable for PTT [15,72]. Yang et al. prepared 2D Nb_2_C MXenedoped BG scaffolds for PTT in the NIR-II biowindow [73]. Nb element in the PTA Nb_2_C could promote the formation of new blood vessels and accelerate the repair of large-scale bone defects [74]. In addition, S-nitrosothiol was also introduced into the composite scaffolds and, by controlling the release of NO from S-nitrosothiol, the anti-tumor ability and bone regeneration effect of the scaffolds were enhanced. Under NIR-II laser irradiation, NO was rapidly released to reach a high concentration, eradicating tumors effectively by inducing DNA damage and inhibiting DNA repair. The tumor was eliminated without recurrence after two weeks of the combined treatment with PTT (1064 nm, 1 w/cm^2^) and gas therapy. In the later stage of treatment, low concentrations of NO contributed to angiogenesis and bone regeneration.

In addition to containing the common elements Si, P, and Ca, other elements such as B, Na, Mg, etc. were also added into BG to improve osteogenic ability [75,76]. Wang et al. first compounded MoS_2_ and PLGA into a thin film and then coated it on the surface of borosilicate BG to prepare integrated treatment scaffolds [77]. The effective control of Mo^4+^ release or retention by PLGA film prolonged and guaranteed the photothermal performance of the scaffolds, as well as reducing the toxicity caused by excess nanomaterials. Due to NIR-triggered photothermal ablation (808 nm, 2 w/cm^2^), the survival rate of tumor cells in vitro was significantly reduced to 15%, compared to 95% in the blank control group.

### 2.3. Other Bioceramic Scaffolds

Akermanite (AKT, Ca_2_MgSi_2_O_7_) is considered a kind of bioceramic with great potential due to its ability to stimulate angiogenesis, good mechanical properties, and controllable degradation rate [78]. The bone regeneration ability of AKT scaffolds could be enhanced after the introduction of coating with 2D borocarbonitrides (BCN), which is mainly due to the following factors [79]: (1) The large specific surface area of BCN improved the adsorption capacity of the scaffolds to bovine serum albumin (BSA) protein; (2) -OH on BCN significantly upregulated the expression of fibronectin (FN); (3) in situ biomineralization properties displayed by BCN; (4) B element in BCN promoted mineralization and osteogenic differentiation. In addition, the strong light absorption of BCN endowed the scaffold with excellent photothermal properties to ablate tumor cells.

Wang et al. realized in situ growth of MoS_2_ nanosheets on the strut surface of AKT scaffolds via a controllable hydrothermal process [80]. Hyperthermia induced by PTT (808 nm, 0.5 w/cm^2^) of scaffolds made 89% of tumor cells necrosis in vivo. Notably, compared to the AKT scaffolds, the composite scaffolds showed a higher ability to enhance the expression of bone-related genes such as osteopontin (OPN), osteocalcin (OCN), and runt-related transcription factor 2 (Runx2) (Figure 8).

Zhuang et al. fabricated Fe-doped, 3D-printed AKT scaffolds for combined photo/magnetothermal therapy and bone regeneration [81]. The combination of magnetothermal therapy (MTT) and PTT (808 nm, 0.75 w/cm^2^) produced higher temperatures with cell viability after photo/magnetothermal treatment was significantly lower than that of PTT or MTT alone in vitro (Figure 9). In addition, Fe-doped AKT scaffolds significantly promoted the proliferation and osteogenic differentiation of rabbit BMSCs compared with AKT scaffolds.

## 3. Polymeric Scaffolds

Polymeric scaffolds are divided into two categories: natural polymers and synthetic polymers, which possess the advantages of biosafety, histocompatibility, and non-toxic degradation products [82]. However, pure polymeric scaffolds are not suitable for use as a matrix material for bone defect repair alone, since the mechanical properties are far from natural bones. In BTE, it is a common and feasible method to fabricate composite scaffolds composed of polymers and bioceramics [83]. The composite scaffolds effectively enhanced tissue cell adhesion, proliferation, and differentiation, and induced bone vascularization to promote bone remodeling [84,85].

### 3.1. Chitosan (CS)

In the past three decades, CS, a natural polymer obtained from chitin, has played an important role in BTE [86]. CS possesses excellent biological activity but no osteoconductivity, and can not meet the requirements of a bone repair material when applied alone [87]. BG/CS composite scaffolds have been shown to support the adhesion, spreading, and proliferation of human BMSCs and promote bone regeneration in vivo [88,89]. Introducing SrFe_12_O_19_ magnetic nanoparticles endowed the scaffolds with excellent antitumor effects. Besides, the magnetic field generated by SrFe_12_O_19_ significantly promoted the osteogenic differentiation of stem cells and bone regeneration by activating the bone morphogenetic protein-2 (BMP-2)/Smad/Runx2 pathway. Ma et al. coated nHA/GO composite particles on CS scaffolds to prepare composite scaffolds with triple functions of PTT, bone regeneration, and hemostasis [90]. The temperature of osteosarcoma tissue increased to 49.9 °C after 150 s of the 808-nm NIR laser irradiation (0.6 w/cm^2^). However, when the tumor was kept at 48 °C, the temperature of the surrounding tissue was much lower (mean 38.5 ± 0.6 °C), which reduced the damage to healthy tissue caused by the high temperature. Intriguingly, NIR significantly promoted the proliferation of cells on scaffolds, suggesting that the appropriate temperature may favor osteogenic differentiation (Figure 10).

Zhao et al. designed multifunctional GdPO_4_/CS/Fe_3_O_4_ scaffolds for breast cancer bone metastases therapy [91]. The hydrated GdPO_4_·H_2_O nanorods were orderly arranged within the CS matrix, which facilitated the formation of new bone. Gd^3+^ released from the scaffolds promoted M2 polarization of macrophages, which significantly stimulated blood vessel formation. In addition, Gd^3+^ also promoted new bone regeneration by activating the BMP-2/Smad/ Runx2 pathway. After two weeks of PTT with 808-nm laser (4.6 w/cm^2^), the tumors of mice became significantly smaller than before.

The derivative product of CS, carboxymethyl chitosan (CMCS), also has good biocompatibility, biodegradability, and osteogenic ability. Compared with CS, it is more water-soluble and therefore has greater application potential [92]. Yao et al. mixed PDA, HA, and CMCS into a slurry to prepare composite scaffolds via 3DP technology [93]. They found that the hyperthermia generated by the photothermal effect not only directly killed bone tumors but also inhibited tumor cell proliferation by inducing apoptosis and inhibiting angiogenesis. The tumor temperature could increase to 50 °C within 1 min under NIR irradiation (808 nm, 1 w/cm^2^), which not only directly killed bone tumors but also inhibited tumor cell proliferation by inducing apoptosis and inhibiting angiogenesis. In addition, PDA was found to promote the osteogenic differentiation of composite scaffolds, possibly because the introduction of PDA increased the surface potential of scaffolds and thus effectively promoted cell diffusion during adhesion and enhanced cell differentiation into osteogenic lineages.

### 3.2. Gelatin (Gel)

Gel is a natural polysaccharide that is partially hydrolyzed from collagen [94]. On account of its cell-adhesive structure, high biocompatibility, and low immunogenicity, the application of Gel has been appropriated from the food industry and cosmetics industry to the biomedical field [95]. Lowering the extraction temperature could obtain Gel with higher mechanical strength; however, it was still not up to the standard in BTE [23,96]. The rapid degradation rate under normal physiological conditions also hinders the application of Gel in BTE [97]. A novel magnetic nanocomposite Gel/AKT scaffold was fabricated to remedy the above deficiencies [98]. The porosity, compressive strength, and elastic modulus of the scaffolds were significantly enhanced after the introduction of PTAs Fe_3_O_4_ and MWNTs (Table 1). In addition to acting as PTAs to eliminate tumor cells, multi-walled carbon nanotubes (MWNTs) could enhance the osteogenic properties of scaffolds due to their excellent mechanical properties, large surface area, easy functionalization ability, and promotion of cell proliferation [99,100].

Osteoinductive and osteoconductive graphene has been shown to improve osteogenesis, in addition to tunable photothermal properties [101,102]. The composite scaffolds prepared by mixing Gel, nHA, CS, and graphene could be rapidly heated to kill tumor cells after NIR laser irradiation [103]. Furthermore, the composite scaffolds could accelerate the proliferation of pre-osteoblastic MC3T3-E1 cells in a mild photothermal environment.

Pan et al. designed bifunctional scaffolds by loading novel MOFs zeolitic imidazolate framework 8 (ZIF8) nanoparticle carriers onto Gel scaffolds [104]. The sintered ZIF8 acquired controllable photothermal properties, which could effectively kill residual bone tumors. After 14 days of PTT (808 nm, 1 w/cm^2^), the tumor volume was significantly reduced compared to the control groups (Figure 11). The mesoporous structure enabled ZIF8 to be loaded with a BMP2 activator phenamil (Phe), and NIR laser irradiation promoted the release of Phe from the scaffolds to enhance BMP2-induced osteogenic differentiation.

### 3.3. Silk Fibroin (SF)

SF is the inner layer of silk, which is mainly derived from Bombyx mori [105]. As terrific mechanical properties, tunable biodegradation rate, and good biocompatibility have been gradually discovered, SF nowadays shows great potential in BTE [106,107]. Miao et al. fabricated PDA functionalized SF scaffolds that could enable the efficient, non-invasive PTT of bone cancer [108]. Interestingly, optical fiber was employed to increase the penetration depth of NIR light. Without using fiber optics, the temperature of the scaffolds was not significantly increased, either when irradiated under pigskin (to simulate human skin) or in mice. Conversely, the temperature after using the fiber optics to penetrate the skin could reach above 40 °C to effectively kill tumor cells (808 nm, 1 w/cm^2^). Inspired by cooking, Ling et al. fabricated hierarchically-porous SF/bacterial cellulose (BC)/Ti_3_C_2_ MXene composite scaffolds using a facile and feasible protein foaming method [109]. Benefiting from the excellent photothermal properties of Ti_3_C_2_ MXene, the composite scaffolds showed an excellent photothermal treatment effect on osteosarcoma, and the percentage of living cells decreased to 30% after the 10 min irradiation by the 808-nm laser (1.5 w/cm^2^). Besides, as MXene’s osteogenic ability was proven, the composite scaffolds produced faster osteogenesis and almost completed the repair of bone defects by new bone tissue compared to controls (Figure 12).

Curcumin (CM), a phenolic pigment extracted from turmeric, not only acted as a chemotherapeutic drug to kill osteosarcoma cells but also enhanced the viability of osteoblasts at appropriate concentrations [110,111]. Meng et al. obtained a novel local drug-delivery system with the abilities of chemo-photothermal synergistic osteosarcoma treatment and bone regeneration by coating PDA on SF/CM scaffolds fabricated by supercritical carbon dioxide (SC-CO_2_) technology [112]. Under the NIR laser irradiation, the release of CM was accelerated by combining it with PTT to synergistically enhance osteosarcoma treatment. The activity of MG-63 cells decreased by only 20.4% after the combined therapy, which was lower than that of chemotherapy (35%) and PTT (37.4%) alone. With the increase in treatment time, the concentration of CM decreased, which could further enhance the osteogenic effect of the scaffolds.

### 3.4. Synthetic Polymers

PCL is a kind of biocompatible aliphatic semi-crystalline polymer that has been approved by the Food and Drug Administration (FDA) [113,114]. The unique nature of slow degradation favors the application of PCL in BTE [115]. Further, PCL is suitable for modification to obtain properties that were not available previously, such as osteoconductivity and osteoinductivity [116]. Wesselsite (SrCuSi_4_O_10_) is a sort of ancient pigment that exhibited strong absorption of NIR-II light after being exfoliated into nanosheets, demonstrating excellent potential in PTT [117]. Yang et al. developed Wesselsite nanosheetfunctionalized PCL scaffolds for NIR-II PTT and enhanced vascularized bone regeneration [118]. The temperature of the tumor site in Saos-2 tumor-bearing mice was as high as 53.4 °C after the NIR-II irradiation (1064 nm, 1 w/cm^2^), and the complete eradication of the tumor could be observed after only four days of treatment. The sustained release of Cu, Si, and Sr ions from composite scaffolds effectively promoted the expression of the osteogenic genes OCN, BMP-2, and Runx2 of rat BMSCs and the angiogenic genes hypoxia-inducible factor-1 (HIF-1α), vascular endothelial growth factor (VEGF), and basic fibroblast growth factor (BFGF) of human umbilical vein endothelial cells (HUVECs) (Figure 13). Quantitative analysis showed that BMD and BV/TV were improved by 1.39- and 1.82-fold, respectively, compared with PCL stents alone.

Egyptian blue (EB, CaCuSi_4_O_10_) is also an ancient pigment with NIR absorption. He et al. obtained EB nanosheets by a facile high-intensity sonication technique and then modified them on the surface of CaCO_3_-PCL scaffolds to construct a multifunctional osteosarcoma treatment platform [119]. EB nanosheets exhibited prominent absorption in the NIR-II biowindow and high PCE, allowing the scaffolds to efficiently eliminate more than 92% of osteosarcoma under 1064-nm laser irradiation (1.5 w/cm^2^) in vitro. The Ca, Cu, and Si components from the scaffolds synergized with each other to jointly enhance bone tissue regeneration. Results from mRNA transcriptome analysis demonstrated that scaffold-based therapy could modulate genes associated with cell death, proliferation, and osteogenesis (Figure 14). PEEK is also an FDA-approved biomaterial with outstanding mechanical properties (elastic modulus) similar to human bone [120]. Zhu et al. developed the first 3D-printed polyetheretherketone (PEEK)-based multi-functional scaffold for challenging bone diseases such as osteosarcoma and osteomyelitis [121]. Along with PTT, the presence of antibiotics and anti-cancer drugs allowed the scaffold to eradicate drug-resistant bacteria and osteosarcoma cancer cells. PTT induced by an 808-nm laser (0.3 w/cm^2^) reduced the tumor weight by 98.5% in vivo in conjunction with chemotherapy. In addition, the combined therapy could mostly eradicate methicillin-resistant *Staphylococcus aureus* and *Escherichia coli*. Coated with bioactive HA, scaffolds promoted serum protein adsorption, cell adhesion, and proliferation, and accordingly resulted in faster healing of bone defects.

## 4. Conclusions and Outlook

More and more studies have been reported recently on bifunctional scaffolds for the treatment of bone tumors. Benefiting from the versatility of PTAs in scaffolds, the functions of scaffolds are not limited to PTT and bone regeneration, but the ability of combination therapy, such as PTT combined with chemotherapy [39,40,112,121], CDT [46], photodynamic therapy (PDT) [47], gas therapy [73], or MTT [81]. Some scaffolds possessed an antibacterial function that effectively prevented the occurrence of intraoperative and postoperative infection [30]. Most of the PTAs introduced previously could release bioactive ions during the degradation process, which enhanced the osteogenic ability of scaffolds by promoting cell adhesion, proliferation, and differentiation, and promoting the formation of new blood vessels. In addition, some studies have pointed out that mild heat stimulation of normal tissue by PTT also promoted the formation of new bone [103,122].

The strategy of modifying osteogenic scaffolds with PTAs not only endowed the scaffolds with antitumor ability but also enhanced osteogenesis. It seems to provide a perfect solution for the treatment of bone tumors; however, there is no effective bone tumor model to simultaneously evaluate the tumor treatment and bone regeneration properties of these scaffolds at present. Bone tumor models cannot be established in rabbits or larger animals due to immune rejection, and the tumor-bearing nude mouse model is currently the most common tumor model. However, the bone size of nude mice is too small to be used effectively in models of bone regeneration defects. The current evaluation of the PTT and bone regeneration properties of bifunctional scaffolds was performed by establishing two independent animal models (ectopic tumor-bearing nude mouse model and bone defect rat/rabbit model). It can be expected that PTT may interact with the process of bone regeneration if an orthotopic bone tumor model is established. Currently, researchers have paid more attention to the promotion effect of NIR irradiation on bone regeneration without considering the influence of the scaffolds on the photothermal properties of PTT. The embedment of the PTAs in the scaffolds usually hinders the heat transfer owing to the presence of the scaffold materials, affecting the PCE. Moreover, to clear the tumor as effectively as possible, the hyperthermia generated by PTT may cause damage to normal cells and tissues near the tumor. Some studies have revealed that PTT and osteogenesis are chronologically independent processes, where the PTT treatment time is shorter than osteogenesis. Even if the short-term PTT kills some osteocytes, long-term bone regeneration is unaffected, as the normal BMSCs in the bone will migrate into the scaffolds. Therefore, the triggering of PTT still needs to be strictly controlled, including power density, time, space, etc., to reduce the damage to the normal tissues. Since bifunctional scaffolds need to exist for a long time in vivo, the toxicity and degradation properties of PTAs are not negligible. There are studies developing a new generation of biomaterials that inherently possess photothermal properties and bone regeneration properties, successfully avoiding the above problems and which opened up a new research direction for the treatment of postoperative bone tumors [123,124].

Currently, research on PTT is mainly focused on the design and development of novel PTAs, such as materials that absorb NIR-II light, which has recently attracted much attention; however, clinical trials of PTT are less reported. Although PTT combined with other therapies can more effectively remove tumors, the necessity, economic benefits, safety, and efficacy of these synergistic therapies need to be discussed in detail.

New bone formation is a complex process that requires the consideration of molecular, biochemical, and cellular factors. In terms of osteogenesis, ideal scaffolds should possess the characteristics of good biocompatibility, vascularization, mechanical properties, osteoconductivity and osteoinductivity. Adjusting the porosity and pore size of scaffolds may affect ossification, angiogenesis, and mechanical and degradation properties. The osteogenic performance of scaffolds can also be improved by compounding bioceramics with polymers. However, at present, the components of bifunctional scaffolds are mostly bioceramics, especially BG, and there are few pieces of research on polymers or composite scaffolds. Moreover, scaffolds can be loaded with bioactive substances to promote bone formation, such as the growth factors BMP-2 and VEGF. Recent studies have shown that successful bone regeneration is based on coordinated interaction between BMSCs and macrophages, and fractures will not heal without the direct involvement of macrophages. However, little attention has been currently paid to the effects of macrophages on bifunctional scaffolds.

At present, the implantation of bifunctional scaffolds into the bone defect site after surgical resectioning of bone tumors provides a promising idea for the postoperative treatment of bone tumors. Under NIR irradiation, the scaffolds heat up rapidly, eliminating nearby tumor cells. The scaffolds slowly degrade over time, helping the bone defect grow new bone. Short-cycle PTT requires PTAs to have high PCE to eradicate tumor cells quickly. Due to a long time for bone repair to take place, PTAs on scaffolds exist in the body for a long time, and it is necessary to reduce their bio-toxicity. PTAs that release bioactive ions have been validated to help with new bone formation and reduce the time to bone regeneration. In addition, it is necessary to develop novel bioscaffolds with stronger osteogenic ability to accelerate the healing of bone defects. To put it succinctly, future research on bifunctional scaffolds should not only focus on toxicity, photothermal effect, and osteogenesis, but also realize accurate time control of short-term tumor elimination and long-term bone repair. Undoubtedly, this requires researchers in the fields of biology, chemistry, physics, medicine, engineering, and more to conduct extensive interdisciplinary research to bring good news to patients as soon as possible.

## Data Availability

Not applicable.

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
