# Peer review of "Two Hawks with One Arrow: A Review on Bifunctional Scaffolds for Photothermal Therapy and Bone Regeneration"

_nanomaterials, 2023, doi:10.3390/nano13030551_

Round 1

Reviewer 1 Report

The authors review the literature on efforts related to bifunctional scaffolds that may be effective for tumor growth inhibition and bone formation. The manuscript discusses a great amount of relevant research but the authors should edit the content to be more precise. They should clearly state in each case whether the data/results they cite come from in vitro or in vivo studies and give more details on the experimental systems used for each of the scaffolds they describe. Thus, it will be much clearer and easier for the reader to compare the scaffolds and draw useful and meaningful conclusions or form new hypotheses/ideas. As it is, with critical information lacking, it is impossible to understand the bifunctional scaffolds' true potential.

The authors should also discuss in more detail any information/insight on how these bifunctional scaffolds would work in vivo. How would they inhibit osteosarcoma cell growth and at the same time promote bone formation?  Although they try to discuss this in the last paragraph of the manuscript, it should be presented in a more concise and detailed manner.

Author Response

To Reviewer #1:

1. The authors review the literature on efforts related to bifunctional scaffolds that may be effective for tumor growth inhibition and bone formation. The manuscript discusses a great amount of relevant research but the authors should edit the content to be more precise. They should clearly state in each case whether the data/results they cite come from in vitro or in vivo studies and give more details on the experimental systems used for each of the scaffolds they describe. Thus, it will be much clearer and easier for the reader to compare the scaffolds and draw useful and meaningful conclusions or form new hypotheses/ideas. As it is, with critical information lacking, it is impossible to understand the bifunctional scaffolds' true potential.

Reply: We greatly appreciate your valuable comments. As suggested, the detailed descriptions for each of the scaffolds were provided to make the bifunctional scaffolds' true potential understood. (see sections 2 and 3 on pages 3-17).

2. The authors should also discuss in more detail any information/insight on how these bifunctional scaffolds would work in vivo. How would they inhibit osteosarcoma cell growth and at the same time promote bone formation?  Although they try to discuss this in the last paragraph of the manuscript, it should be presented in a more concise and detailed manner.

Reply: This is a really good comment. According to your suggestion, we provided more information on how these bifunctional scaffolds would work in vivo and how would they inhibit osteosarcoma cell growth and at the same time promote bone formation in paragraph 2 of section 4. Also, the respective effects of PTT of PTAs and bone regeneration of scaffolds have been described in section 1. However, there is currently no valid bone tumor model to evaluate the tumor therapy and bone regeneration properties of these bifunctional scaffolds simultaneously. (see section1and 4, pages1-2, 19).

Reviewer 2 Report

The manuscript makes a comprehensive review of the most important scientific research published up to this date related to the use of scaffolds in photothermal therapy applications in bone cancer and their secondary role as a long term promoter for bone regeneration. Stresses the complexity of this process, the several solutions used and the necessity for further research and investment in this field, to overcome some of the problems that the current scaffolds have. This is a very good synthesis manuscript that will be useful for the researchers in the field and for the new student that is starting now. Recommend for publication in current form.

Author Response

To Reviewer #2:

The manuscript makes a comprehensive review of the most important scientific research published up to this date related to the use of scaffolds in photothermal therapy applications in bone cancer and their secondary role as a long term promoter for bone regeneration. Stresses the complexity of this process, the several solutions used and the necessity for further research and investment in this field, to overcome some of the problems that the current scaffolds have. This is a very good synthesis manuscript that will be useful for the researchers in the field and for the new student that is starting now. Recommend for publication in current form.

Reply: Thanks a lot for your valuable comments.

Reviewer 3 Report

This manuscript is well written but the Authors should include at least one figure in the manuscript that shows the Author's perspective on the subject.

Additional comment:

Tthe authors have prepared a review paper on the problem of bifunctional scaffolds for photothermal therapy and bone regeneration. All problems were presented in a logical and clear way. The conclusions are consistent with the presented evidence. References are appropriate. However, authors should add to the manuscript at least one figure of their own authorship that would show the author's conclusions of this problem.

Author Response

To Reviewer #3:

This manuscript is well written but the Authors should include at least one figure in the manuscript that shows the Author's perspective on the subject.

Reply: Thank you very much for your good suggestion. A picture has been added as scheme 1 in the introduction on page 2.

Reviewer 4 Report

The bifunctional scaffolds simultaneously ablate bone tumors and generate new tissues at the bone defects, which is a very interesting project. This review summarize the recent application progress of various bifunctional scaffolds and puts forward some practical constraints and future perspectives on bifunctional scaffolds for tumor therapy and bone regeneration. This is a good review, here are a few comments for revision:

1)When the PTT agents like Au, CuS NPs are embedded into the scaffolds, will this block the heat transmission for PTT and how is this affect the PTT efficiency?

2)CuS is a good PTT agent and Copper can induce Fenton reaction for cancer treatment, plus CuS can be sued for both PTT and PDT, please add CuS as its has so many merits, the authors may see these papers for more information - Nanomedicine (London), 2010, 5(8): 1161-1171;J. Biomed. Nanotechnol. 2012, 8 (6), 883-890 and Photodiagnosis and Photodynamic Therapy, 2017, 19:5-14

Author Response

To Reviewer #4:

1. When the PTT agents like Au, CuS NPs are embedded into the scaffolds, will this block the heat transmission for PTT and how is this affect the PTT efficiency?

Reply: Your valuable questions are greatly appreciated. Currently, most studies integrated PTAs into scaffolds for the purpose to endow scaffolds with photothermal capabilities. More attentions have been majorly paid to the promotion effect of the NIR irradiation on bone regeneration but not the influence of the presence of scaffolds on the photothermal properties of PTT. Although the photothermal performance of bifunctional scaffolds was explored in all the articles, rare description in the effect of scaffolds on photothermal conversion efficiency was reported compared with the single PTAs, which needs to be well explored in the future. So this issue was included in the outlook of bifunctional scaffolds for PTT and bone regeneration. (see section 4, page 19)

2. CuS is a good PTT agent and Copper can induce Fenton reaction for cancer treatment, plus CuS can be sued for both PTT and PDT, please add CuS as its has so many merits, the authors may see these papers for more information - Nanomedicine (London), 2010, 5(8): 1161-1171;J. Biomed. Nanotechnol. 2012, 8 (6), 883-890 and Photodiagnosis and Photodynamic Therapy, 2017, 19:5-14

Reply: According to your valuable suggestion, the introduction of CuS has been  added in section 2.2.2 by citing the mentioned papers as Refs. 52-54. (see page 7 and references section)

Round 2

Reviewer 1 Report

The authors have significantly improved the manuscript. They could incorporate the added Scheme 1 as Figure 1 and renumber all figures.